# Analytical Modeling of In-Process Temperature in Powder Bed Additive Manufacturing Considering Laser Power Absorption, Latent Heat, Scanning Strategy, and Powder Packing

**DOI:** 10.3390/ma12050808

**Published:** 2019-03-08

**Authors:** Jinqiang Ning, Daniel E. Sievers, Hamid Garmestani, Steven Y. Liang

**Affiliations:** 1Georgia Institute of Technology, George W. Woodruff School of Mechanical Engineering, 801 Ferst Drive, Atlanta, GA 30332-0405, USA; 2The Boeing Company, 499 Boeing Boulevard, Huntsville, AL 35824, USA; daniel.e.sievers@boeing.com; 3Georgia Institute of Technology, School of Materials Science and Engineering, 771 Ferst Drive NW, Atlanta, GA 30332, USA; hamid.garmestani@mse.gatech.edu

**Keywords:** in-process temperature in MPBAM, analytical modeling, high computational efficiency, molten pool evolution, laser power absorption, latent heat, scanning strategy, powder packing

## Abstract

Temperature distribution gradient in metal powder bed additive manufacturing (MPBAM) directly controls the mechanical properties and dimensional accuracy of the build part. Experimental approach and numerical modeling approach for temperature in MPBAM are limited by the restricted accessibility and high computational cost, respectively. Analytical models were reported with high computational efficiency, but the developed models employed a moving coordinate and semi-infinite medium assumption, which neglected the part dimensions, and thus reduced their usefulness in real applications. This paper investigates the in-process temperature in MPBAM through analytical modeling using a stationary coordinate with an origin at the part boundary (absolute coordinate). Analytical solutions are developed for temperature prediction of single-track scan and multi-track scans considering scanning strategy. Inconel 625 is chosen to test the proposed model. Laser power absorption is inversely identified with the prediction of molten pool dimensions. Latent heat is considered using the heat integration method. The molten pool evolution is investigated with respect to scanning time. The stabilized temperatures in the single-track scan and bidirectional scans are predicted under various process conditions. Close agreements are observed upon validation to the experimental values in the literature. Furthermore, a positive relationship between molten pool dimensions and powder packing porosity was observed through sensitivity analysis. With benefits of the absolute coordinate, and high computational efficiency, the presented model can predict the temperature for a dimensional part during MPBAM, which can be used to further investigate residual stress and distortion in real applications.

## 1. Introduction

Metal powder bed additive manufacturing (MPBAM), alternatively named powder bed fusion (PBF), is one of the widely used additive manufacturing processes, in which high-density laser power is used to selectively melt and fuse powders to the build part in a layer by layer manner. MPBAM is capable of producing geometrically complex parts with effective cost [1]. The large temperature gradient in MPBAM is frequently observed due to the repeatedly rapid heat and solidification, which are detrimental to the quality of the produced part by causing defects, such as cracking [2], undesired residual stress [3], and part distortion [4,5], and thus alternating the part’s mechanical properties and functionality [6,7,8]. Therefore, the monitor or prediction of the temperature distribution in MPBAM is needed.

In situ temperature measurement is difficult and inconvenient due to the restricted accessibility under elevated temperature conditions [9,10]. Non-contact and contact techniques are employed for temperature measurements in the additive manufacturing (AM) process. Non-contact thermal photographic techniques, such as an infrared (IR) pyrometer and an IR camera, can only measure temperatures on the exposed surfaces [11,12]. Contact techniques, such as an embedded thermocouple, can only measure temperatures inside the substrate rather than the build [13,14]. In fact, the temperatures inside the build have a direct influence on the quality of the produced part. More in situ measurement techniques in the AM process can be found in the review literature [15,16]. In addition, molten pool measurement using an optical microscope based on the solidified microstructure is a post-processing technique for thermal analysis, which requires extensive experimental work for sample preparation, such as cutting, polishing, and etching [17,18].

Numerical models based on finite element method (FEM) were developed to address the difficulty and inconvenience in monitoring the additive manufacturing (AM) process. Roberts developed a FEM model to predict the temperature distribution in selective laser melting (SLM) of Ti-6Al-4V involving multiple layers with an element birth and death technique [19]. Similar numerical models were developed with different types of heat sources considering the absorptivity and the different shape of the heat source [20,21]. Fu et al. developed a FEM model to predict the temperature distribution in the SLM of Ti-6Al-4V using powder material properties and bulk material properties. Improved prediction accuracy was reported using powder material properties upon validation to experimental molten pool dimensions [22]. FEM models were also developed for temperature prediction in SLM of various materials, including aluminum, titanium, stainless steel, and Inconel alloy [23,24,25,26]. Criales et al. investigated the sensitivity of material properties and process parameters in the prediction of SLM of Inconel 625 [27]. Papadakis et al. developed a computational reduced model for prediction in SLM with improved computational efficiency, in which the heat in each scanning vector was quantified as an input [28]. The recent numerical models have considered the influence of powder packing and molten pool dynamics in the temperature prediction, which allowed the investigation of defects in the produced parts [29,30,31]. Qi et al. and Kolossov et al. developed FEM models to predict the temperature distribution in a coaxial laser cladding process and selective laser sintering (SLS) process, respectively [32,33]. Residual stress and part distortion in the AM process were also investigated through numerical modeling [34,35,36]. The detailed discussion of FEM models is out of the scope of this work. More details can be found in the review literature [37,38,39]. Although FEM models have made considerable progress in predicting the AM processes, the high computational cost is still the major drawback.

Analytical models have demonstrated their high computational efficiency, high prediction accuracy, and broad applicability in predicting the manufacturing processes, and in the inverse determination of the material constants [40,41]. To overcome the drawback in computational efficiency, analytical models were also developed to predict the AM process. Peyre et al. developed a semi-analytical model to predict the temperature in direct metal deposition (DMD), in which an analytical model and a FEM model were used to characterize the deposition geometry and temperature distribution, respectively [42]. Yang et al. developed another semi-analytical model to predict the temperature in SLM, in which an analytical model and a FEM model were used to characterize the moving heat source and impose heat transfer boundary condition, respectively [43]. Van Elsen et al. summarized three moving heat source solutions based on the coordinate with a moving origin at the heat source location, namely moving point heat source, moving semi-ellipsoidal heat source, and moving uniform heat source [44]. Isotropic and homogeneous material and semi-infinite medium were assumed in those models. The aforementioned heat source solutions were originally developed by Carslaw and Jaeger [45]. The moving point heat source was further developed with consideration of heat source shape for temperature prediction in SLM [46]. Rosenthal developed a moving line heat source solution to predict the temperature in welding for an infinite thin plate [47]. The line heat source solution was adopted to predict the temperature in coaxial laser cladding, in which single-track scan prediction and semi-infinite medium assumption were enforced [48]. Tan et al. further developed the line heat source solution by transforming the moving coordinate to the absolute coordinate for consideration of the part dimensions. The final solution was constructed from the superposition of the actual heat source and two image heat sources [49]. This solution is limited only for continuous single-track scans. However, the FEM models used in semi-analytical models prevent optimized computational efficiency; most developed analytical models employed a moving coordinate and assumed a semi-infinite medium and steady-state condition through the process. Those assumptions reduced the usefulness of the developed model in real applications because of the lack of time-dependence and location-dependence related to part dimensions. Moreover, Green’s Function has been widely used for temperature and stress predictions of the bounded medium due to thermal and mechanical loads, but the high mathematical complexity resulted in an unoptimized computational efficiency [45,50,51]. The temperatures in a dimensional part cannot be accurately and efficiently predicted with the developed analytical models for multi-track scans in MPBAM.

This paper presents an analytical model to predict in-process temperatures in MPBAM based on a stationary coordinate, whose origin is located at the part boundary (absolute coordinate), with consideration of the laser power absorption, latent heat, scanning strategy, and powder packing porosity. Analytical solutions are developed to predict temperatures in single-track scans and multi-track scans under various process conditions. Molten pool dimensions are then obtained by comparing the predicted temperatures to the material melting temperature. Inconel 625 was chosen to test the presented models with validation to the molten pool dimensions in the literature, which were measured based on the solidification microstructure [18]. The following tasks were performed in this study: (1) to inversely determine the laser power absorption with comparison between predicted molten pool dimensions and experimental values using trial-and-error method; (2) to investigate the molten pool evolution with respect to scanning time; (3) to predict the stabilized molten pool dimensions in single-track scans and bidirectional scans under various process conditions, and then perform validations against experimental values; (4) to record and present the computation time; (5) to investigate the influence of powder packing porosity on the predicted temperatures through sensitivity analyses.

The employed absolute coordinate and time consideration allows the temperature prediction for a dimensional part, which significantly improves the usefulness of the developed model in real applications. For comparison, other analytical models assume steady-state condition, and thus they are not applicable for temperature prediction at the beginning of the scan, where the part boundary is located at. The in-process temperature analysis allows the investigation of molten pool evolution, specifically the growth and stabilization of the molten pool. The high computational efficiency allows the inverse determination of laser powder absorption based on the experimental measurements.

## 2. Methodology

In this work, the in-process temperature in MPBAM is predicted through analytical modeling based on the absolute coordinate. The heat balance governing equation is expressed as
(1)$$\frac{\partial \rho u}{\partial t}+\frac{\partial \rho HV}{\partial x}=\;\nabla \cdot \left(k\nabla T\right)+\dot{q}$$
where *u* is internal energy, *H* is enthalpy, *ρ* is density, *k* is conductivity, and $\dot{q}$ is a volumetric heat source, *t* is time, *x* is distance, *V* is heat source moving speed, and *T* is temperature. With *V* = 0, and $du=C_{p}dT$, where $C_{p}$ is specific heat, the heat balance equation becomes the heat conduction equation expressed as
(2)$$\frac{\partial^{2}T}{\partial x^{2}}+\frac{\partial^{2}T}{\partial y^{2}}+\frac{\partial^{2}T}{\partial z^{2}}=\;\frac{1}{\kappa }\frac{\partial T}{\partial t}+\dot{q}$$
where κ is thermal diffusivity ($\kappa =k/\rho C_{p}$), x,y,z denote three mutually perpendicular directions in the absolute coordinate.

A point heat source solution is developed by Carslaw and Jaeger [45] with satisfaction of the heat conduction equation as the following.
(3)$$\theta \left(x,y,z\right)=\;\frac{Q}{8{\left(\pi \kappa t\right)}^{\frac{3}{2}}}\exp\left[-\frac{x^{2}+y^{2}+z^{2}}{4\kappa t}\right]$$
where Q is the amount of heat, θ is temperature change ($\theta =T-T_{0}$).

The moving point heat source solution at the current time (t) and location (X−V(t−t′),Y,Z) due to the heat input at the previous time $\left(t^{\prime }\right)$ was derived from the point heat source solution as the following.
(4)$$\theta \left(x,\;y,z,\;t\right)=\frac{Pdt\prime }{8\rho c{\left[\pi \kappa \left(t-t\prime \right)\right]}^{\frac{3}{2}}}\exp\left[-\frac{{\left\{x-V\left(t-t^{\prime }\right)\right\}}^{2}+y^{2}+z^{2}}{4\kappa \left(t-t^{\prime }\right)}\right]$$
where P, V are laser power and laser scanning velocity, respectively, *x*-direction is assumed to be laser scanning direction, *y*-direction is assumed to the hatch direction, as illustrated in Figure 1a.

The 3D in-process temperature can be calculated by integrating the temperature solution with a time range from 0 to *t*. With the consideration of laser power absorption (*η*), the temperature solution becomes the following.
(5)$$\theta \left(x,y,z,t\right)=\;\;\frac{P\eta }{8\rho C_{p}{\left(\pi \kappa \right)}^{\frac{3}{2}}}\int_{0}^{t}\frac{\exp\left[-\frac{{\left(x-V\left(t-t^{\prime }\right)\right)}^{2}+y^{2}+z^{2}}{4\kappa \left(t-t^{\prime }\right)}\right]}{{\left(t-t^{\prime }\right)}^{\frac{3}{2}}}dt^{\prime }$$

The temperature solution can be further derived by integrating $t^{\prime }$ from 0 to *t* as the following.
(6)$$\theta \left(x,\;y,z,\;t\right)=\frac{P\eta }{2Rk_{t}\pi^{\frac{3}{2}}}exp\left(\frac{Vx}{2\kappa }\right)\int_{\frac{R}{2\sqrt{\pi t}}}^{\infty }exp\left[-\xi^{2}-\left(\frac{V^{2}R^{2}}{16\kappa^{2}\xi^{2}}\right)\right]d\xi$$
where $R^{2}=\;x^{2}+y^{2}+z^{2}$, t is the current time, t′ is previous time, x, y, z are the corresponding distances from the laser source, ξ is an integration variable which leads to the concise expression.

The 2D in-process temperatures in the single-track scan can be calculated by setting *y*-direction distance as zero.
(7)$$\theta \left(x,0,z,t\right)=\;\;\frac{P\eta }{8\rho C_{p}{\left(\pi \kappa \right)}^{\frac{3}{2}}}\int_{0}^{t}\frac{\exp\left[-\frac{{\left(x-V\left(t-t^{\prime }\right)\right)}^{2}+z^{2}}{4\kappa \left(t-t^{\prime }\right)}\right]}{{\left(t-t^{\prime }\right)}^{\frac{3}{2}}}dt^{\prime }$$

The 2D in-process temperatures in the multi-track scan with hatch space (h) is calculated by combining temperature at the current track location due to the energy input from the previous track $\left(\theta_{1}\right)$ and the temperature due to the added energy input from the current track $\left(\theta_{2}\right)$. The in-process temperatures can be expressed as the following.
(8)$$\theta \left(x,\;0,z,\;t\right)=\;\theta_{1}+\theta_{2}$$
(9)$$\theta_{1}\left(x,h,z,t_{1}\right)=\;\frac{P\eta }{2Rk_{t}\pi^{\frac{3}{2}}}\exp\left(\frac{Vx}{2\kappa }\right)\int_{\frac{R}{2\sqrt{\pi t}}}^{\infty }\exp\left[-\xi^{2}-\left(\frac{V^{2}R^{2}}{16\kappa^{2}\xi^{2}}\right)\right]d\xi$$
(10)$$\theta_{2}\left(x,0,z,t\right)=\;\frac{P}{8\rho C_{p}{\left(\pi \kappa \right)}^{\frac{3}{2}}}\int_{t1}^{t}\frac{exp\left[-\frac{{\left(x-V\left(t-t^{\prime }\right)\right)}^{2}+y^{2}+z^{2}}{4\kappa \left(t-t^{\prime }\right)}\right]}{{\left(t-t^{\prime }\right)}^{\frac{3}{2}}}dt^{\prime }$$
(11)$$\theta_{2}\left(x,0,z,t\right)=\;\frac{P}{8\rho C_{p}{\left(\pi \kappa \right)}^{\frac{3}{2}}}\int_{t1}^{t}\frac{exp\left[-\frac{{\left(x+V\left(t-t^{\prime }\right)\right)}^{2}+y^{2}+z^{2}}{4\kappa \left(t-t^{\prime }\right)}\right]}{{\left(t-t^{\prime }\right)}^{\frac{3}{2}}}dt^{\prime }$$
where a positive scanning velocity is used for unidirectional scans, as in Equation (9); a negative scanning velocity is used for bidirectional scans, as in Equation (10). The unidirectional scans and bidirectional scans are illustrated in Figure 1b. Additionally, $t_{1}$ is the required time for completing previous track in two consecutive tracks and t is the current time in two consecutive tracks. Then, the transient temperature in multi-track scans can be calculated in the same manner.

In addition, the latent heat is considered using the heat integration method, in which the temperature of the molten pool material is lowered by an amount as the following, due to the phase transformation [44].
(12)$$\Delta T=\;H_{f}/C_{p}$$

The influence of powder packing porosity on the predicted temperatures is investigated using effective material properties as the following.
(13)$$\rho_{e}={\left(1-\tau \right)}^{\gamma }\rho$$
(14)$$k_{e}={\left(1-\tau \right)}^{\beta }k$$
where subscript *e* denotes effective values, τ is powder packing porosity, γ, β are coefficients that can be taken as 1, as suggested by Criales et al. [27].

## 3. Results and Discussion

To investigate the accuracy and effectiveness of the presented model, temperatures in MPBAM of Inconel 625 were predicted under different process conditions in single-track scans and bidirectional scans. Six different process conditions were used for single-track scans and bidirectional scans separately, as given in Table 1. Molten pool dimensions were then obtained from the comparison between the predicted temperatures and material melting temperature, as illustrated in Figure 2. The determined molten pool dimensions were validated by the experimental measurements in the literature, in which an EOS M2070 machine (ASTM International, West Conshohocken, PA, USA) was used in the MPBAM process with gas atomized powders, and a digital optical microscope was used to measure the molten pool dimensions based on the solidification microstructure. The machining uses a single-mode, continuous wave ytterbium fiber laser under nitrogen gas ambiance. The gas automized powder has an average size of 35 μm with powder size distribution of D60% = 29.4 μm, D10% = 13.5 μm, D90% = 43 μm (D denotes powder diameter). More details of experimental design can be found in the reference [18].

The thermophysical material properties of Inconel 625 alloy are given in Table 2. Laser power absorption is affected by laser and powder materials, such as laser wavelength, powder material properties, powder packing-related surface roughness, laser-workpiece standoff distance, etc. [52,53]. Therefore, a given laser power absorption should be valid only for a specific experimental setup. Since the laser power absorption has not been measured and reported in the literature, the absorption was inversely determined using the presented model by minimizing the difference between predicted molten pool depth and experimental molten pool depth under test 6 process conditions. Inverse analysis has been widely used in the determination of materials constants and physical parameters [54]. The trial-and-error method was employed with varying absorption values. The laser absorption was determined as 40% for the current study, as shown in Figure 3. A linear relationship was observed with underestimated absorptions. A non-linear relationship was observed with overestimated absorptions, which might be caused by the more significant influence of latent heat on the overestimated temperature zone. The determined laser absorption (40%) was used in the following studies in this work. The simple calculations in the presented solutions without mesh and iteration, which FEM models rely on, lead to the high computational efficiency of the developed model. Computational time and implementation details were provided for the investigation of the computational efficiency of the presented model. A MATLAB program was used to implement the predictions using the presented model on a personal computer running at 2.8 GHz. The computation time under each process condition was 27 s less for an area near a heat source, which has a length (*x* = 1 mm) with 5 μm increments and a height (*z* = 0.3 mm) with 1 μm increments. Other related variables of molten pool length, molten pool depth, and corresponding prediction error are given in Table 3.

To investigate the evolution of molten pool during the MPBAM, the in-process temperatures were predicted in a single-track scan under test 6 process conditions. Predicted temperature profiles on the top view and the cross-sectional view along the laser scanning direction were illustrated in Figure 4.

The molten pool growth for the stabilization process was investigated with a time interval from 0.001 ms to 10 ms, which corresponded to a distance interval from 0.0008 mm to 8 mm. The trends of molten pool length, molten pool width, and molten pool depth over time were predicted using the presented model, as shown in Figure 5a. The molten pool became stable after 1 ms. The trend of molten pool volume over time is shown in Figure 5b, which was calculated as the following [22]. All associated data is given in Table A1 in the Appendix A.
(15)$$Vol=\;\frac{\pi DLW}{6}$$

Moreover, the molten pool growth and stabilization process in the second track of two consecutive bidirectional scans was predicted, as shown in Figure 6. The molten pool in the second track of bidirectional scans became stable after 0.5 ms, which was faster than that in the single track scan (1 ms). In addition, the larger molten pool length and width were observed in the bidirectional scan after stabilization because of the heat affected zone due to the previous track.

The stabilized melting depth values under various process conditions were predicted and validated to the experimental values in the literature [18]. The temperature distributions along the depth (*z*-direction) in a single-track scan and bidirectional scans under test 6 process conditions were predicted, as shown in Figure 7. The maximum temperature in a bidirectional scan is higher than that in the single-track scan. The constant temperature region in both scans is due to the consideration of latent heat, in which phase transformation takes place rather than temperature increase with continuous heat input. A similar trend was also reported in the literature [19,32,44]. Close agreements were observed upon validation to experimental values under various process conditions for single-track scans and bidirectional scans, as shown in Figure 8a,b, respectively. The deviations between predictions and experiments might be caused by the molten pool shrinkage [55], which was assumed to be negligible. The predicted melt length, melt depth, and computation time are given in Table 4. The average computation time for 2D temperature prediction in single-track scans was 19.44 s; the average computation time for 2D temperature prediction in bidirectional scans was 88.17 s. However, a 3D temperature prediction requires much longer computational time, especially in bidirectional scans, because a large number of 2D temperature profiles need to be calculated iteratively with consideration of the influence of the heat affected zone produced by the previous scans.

Moreover, the influence of powder packing porosity on predicted temperature was investigated by sensitivity analyses for single-track tests and bidirectional tests separately. The packing porosity was deliberately given as 0%, 20%, 40%, and 60% in an increasing trend. The simple fraction models (in Equations (13) and (14)) were employed to correlate between material properties and powder packing. A similar method has been employed for sensitivity analysis, as reported in the literature [27]. It should be noted that the fraction models calculate materials’ properties considering the porosity of packed powder rather than the porosity of the build. The build porosity is affected by powder packing and process conditions. Positive correlations were observed between the packing porosity and melt depth, as illustrated in Figure 9. Therefore, the increase in powder packing porosity leads to an increase in molten pool dimensions, and vice versa. This finding confirms the instinctive trend because larger porosity leads to a lower value for thermal conductivity, which prevents energy being dissipated into the build.

This work investigated the molten pool evolution, including the growth and stabilization of the molten pool using the presented temperature model with analytical solutions. The recorded and presented computation time demonstrated the high computational efficiency of the presented model. With the absolute coordinate and the extended predictive capability for multi-track scans, the temperature distribution can now be predicted for a dimensional part, which significantly improved the usefulness of the presented model in real applications. The high computational efficiency also allows the process-parameters planning for desired temperature conditions. The predicted temperature can be used to further investigate residual stress and part distortion [56,57]. It should be noted that the cooling state after laser turning off and cooling-associated molten pool shrinkage were not considered in this study. The limitations of the presented model are the following: the limited predictive capability for single layer scanning; the assumption of temperature-independent material properties; the assumption of simplified point heat source. The cooling state, extended predictive capability for multiple layer scans, temperature-dependent material properties, and three-dimensional heat source should be considered in future works.

## 4. Conclusions

This work investigates the in-process temperature in MPBAM by further developing a moving point heat source solution, as originally proposed by Carslaw and Jaeger [45], to extend its applicability from a single-track scan to multi-track scans with consideration of laser power absorption, latent heat, scanning strategy, and powder packing porosity. The absolute coordinate allows the temperature prediction for a dimensional part with consideration of time-dependence and location-dependence, with respect to the beginning time and location of MPBAM. The laser power absorption was inversely determined using trial and error method based on experimental measurement of molten pool dimensions. The latent heat was considered by heat integration method. The extended applicability and the absolute coordinate significantly increase the usefulness of the developed model in real applications. For comparison, other analytical models predicted temperature distribution in single-track scans with the semi-infinite medium assumption, in which a moving coordinate was defined with the origin located at the laser heat source. To test the presented model, Inconel 625 alloy was chosen to predict the temperatures in MPBAM under various process condition. Good agreements were observed upon validation against experimental values in the literature. The molten pool evolution was investigated to demonstrate the molten pool growth and stabilization during the scanning process. It should be noted that the cooling state in the post-process has not been considered in the presented model. A positive correlation between molten pool dimensions (temperature distribution) and powder packing porosity was revealed through sensitivity analyses using the presented model. With the benefits of extended applicability for multi-track scans, the absolute coordinate, and the high computational efficiency, the developed model can be used for temperature investigation in real applications.

## Figures and Tables

**Figure 1 materials-12-00808-f001:**
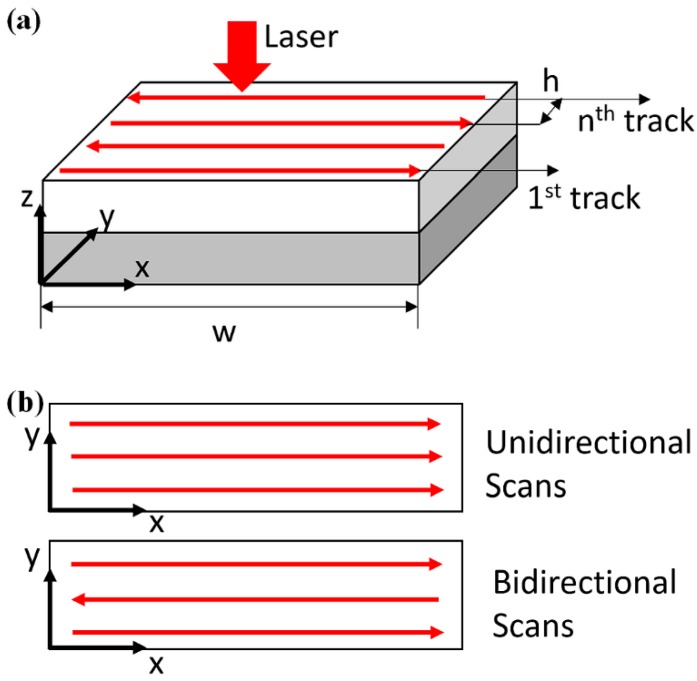
(**a**) Schematic view of the additive manufacturing process, where *w*, *h* denote track length and hatch space, respectively; (**b**) schematic view of unidirectional scans and bidirectional scans.

**Figure 2 materials-12-00808-f002:**
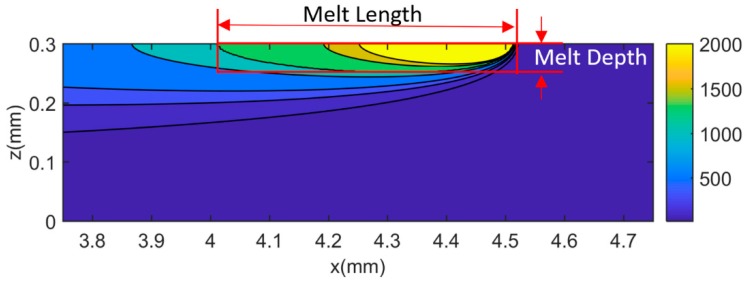
Predicted temperature distribution and molten pool geometry under test 6 process conditions, using 40% absorption. It should be noted that the color bar values correspond to the temperature contours.

**Figure 3 materials-12-00808-f003:**
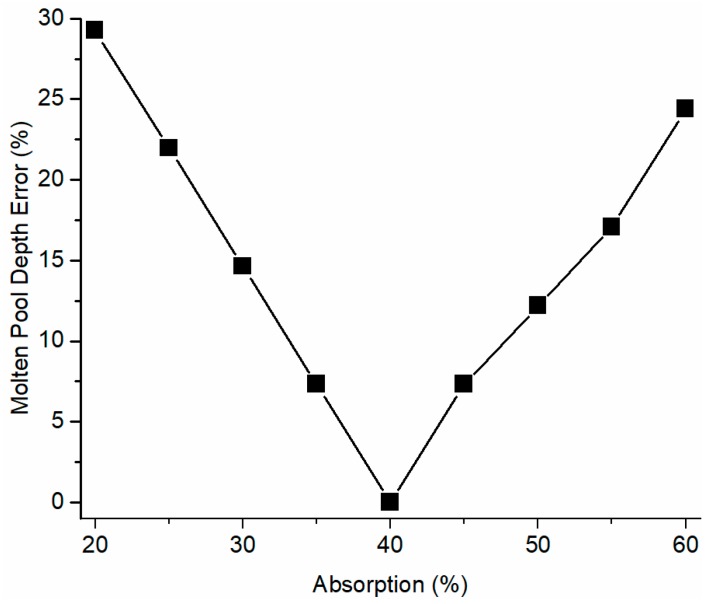
Inverse determination of laser absorption based on melt depth prediction using the trial-and-error method under test 6 process condition.

**Figure 4 materials-12-00808-f004:**
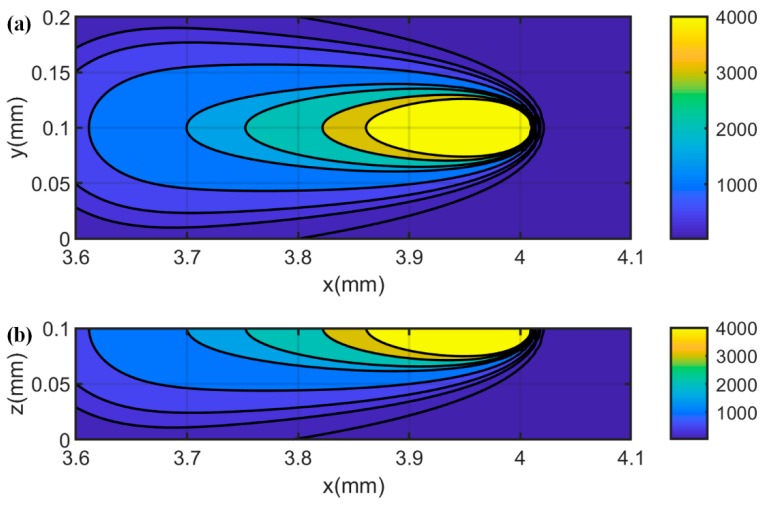
Predicted temperature distribution at *t* = 5 ms (laser location *x* = 4 mm, *y* = 0.1 mm) in single track under test 6 process conditions using 40% absorption. (**a**) Top view; (**b**) cross-sectional view at laser scanning location. It should be noted that the color bar values are corresponding to the temperature contours.

**Figure 5 materials-12-00808-f005:**
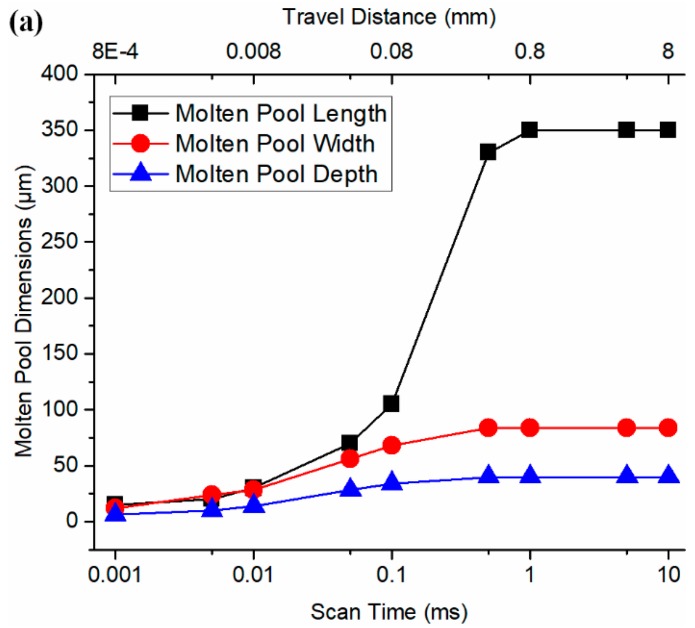
The growth and stabilization of the predicted molten pool in single track scan using the determined absorption of 40% in terms of (**a**) molten pool length, molten pool width, and molten pool depth, and (**b**) molten pool volume.

**Figure 6 materials-12-00808-f006:**
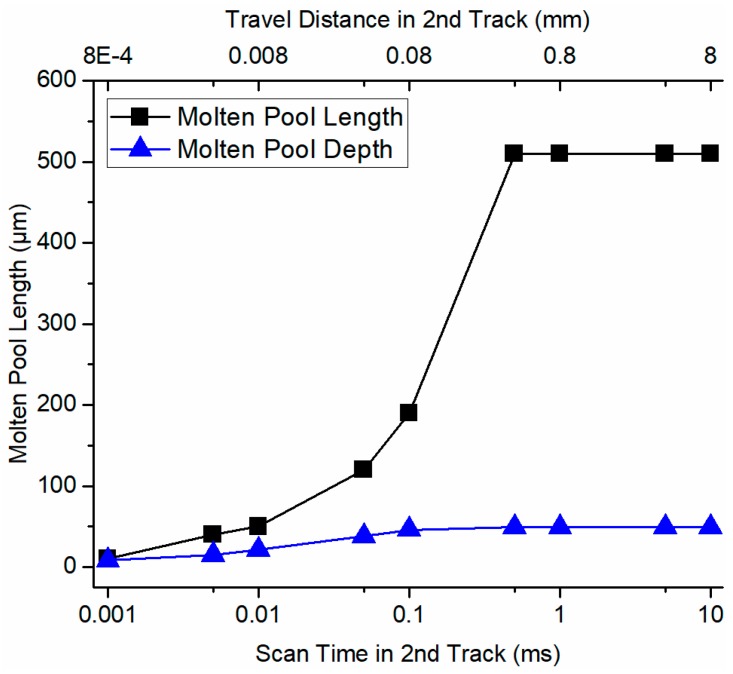
The growth and stabilization of predicted molten pool in the second track of bidirectional scans using the determined absorption of 40% in terms of molten pool length and molten pool depth.

**Figure 7 materials-12-00808-f007:**
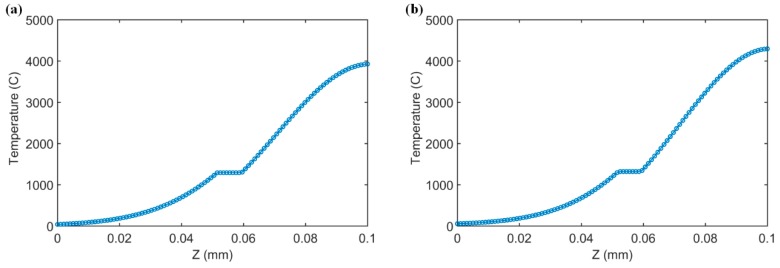
Stabilized temperature distribution from prediction along depths (z-direction) at laser scan locations (*x* = 4 mm, *y* = 0.1 mm) for (**a**) single-track scan and (**b**) bidirectional scan under test 6 conditions. The minimum temperatures are at room temperature (20 °C). The determined absorption of 40% was used in the predictions.

**Figure 8 materials-12-00808-f008:**
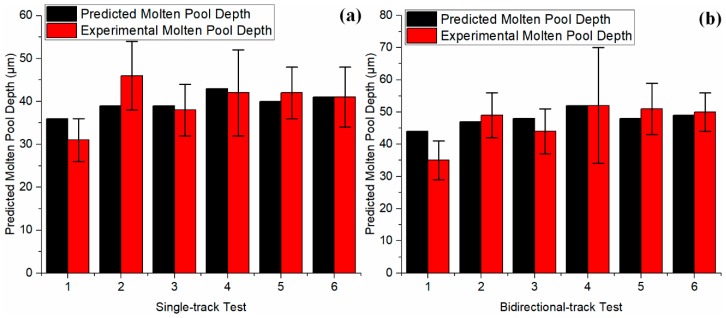
Validation of predicted molten pool depth in (**a**) a single track scan and (**b**) a bidirectional scan after stabilization. Black color represents predicted melt depth. Red color represents experimental melt depth based on solidified microstructure. The determined absorption of 40% was used in the predictions.

**Figure 9 materials-12-00808-f009:**
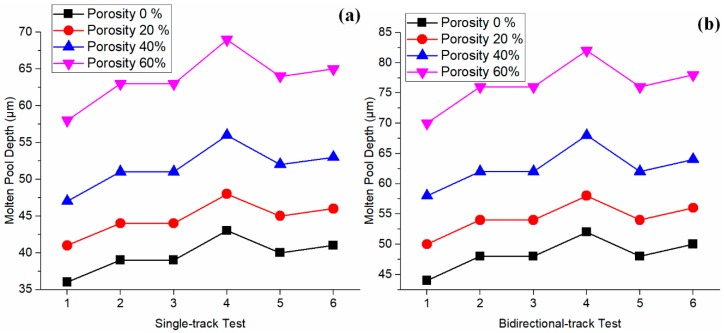
Influence of powder packing porosity on temperature distribution through sensitivity analyses in (**a**) single track scans and (**b**) bidirectional scans. Note: the presented results are obtained from prediction using the simple fraction models with the determined absorption of 40%.

**Table 1 materials-12-00808-t001:** Process conditions for single-track scans and bidirectional scans in SLM of Inconel 625 [18]. The layer thickness is 20 μm.

Test	Laser Powder *P* (W)	Scanning Velocity *V* (mm/s)	Hatch Space *h* (mm)
1	169	875	0.1
2	195	875	0.1
3	182	800	0.1
4	195	725	0.1
5	169	725	0.1
6	195	800	0.1

**Table 2 materials-12-00808-t002:** Material properties of Inconel 625 and inversely determined laser absorption ($T_{0}=20\;{}^{\circ}C$) [14,27].

Density *ρ* (kg/m^3^)	Thermal Conductivity *k* (W/m–°C)	Specific Heat *C_p_* (J/kg–°C)	Solidus Temperature *T_s_* (°C)	Liquidus Temperature *T_L_* (°C)	Latent Heat *H_f_* (J/kg)	Absorption *η* (%)
8840	9.8	410	1290	1350	227,000	40

**Table 3 materials-12-00808-t003:** Inverse determination of laser power absorption in this study.

Absorption η (%)	Molten Pool Length L (μm)	Molten Pool Depth D (μm)	Molten Pool Depth Error Error (%)	Computation Time $t_{c}\;(s)$
20	180	29	29.27	26.11
25	225	32	21.95	24.46
30	270	35	14.63	23.06
35	310	38	7.32	22.95
40	355	41	0.00	23.04
45	395	44	7.32	23.23
50	440	46	12.20	22.95
55	480	48	17.07	23.10
60	525	51	24.39	23.20

**Table 4 materials-12-00808-t004:** Predicted molten pool dimensions for single-track scans and bidirectional scan under steady-state conditions at *x* = 4 mm, *y* = 0.1 mm, with 2 μm increments in x and z directions.

Single-Track Test	Molten Pool Length L (μm)	Molten Pool Depth D (μm)	$\mathrm{Computation}\;\mathrm{Time}\;t_{c}\;(s)$	Bidirectional Test	Molten Pool Length L (μm)	Molten Pool Depth D (μm)	$\mathrm{Computation}\;\mathrm{Time}\;t_{c}\;(s)$
1	310	36	19.60	1	450	44	87.78
2	360	39	18.90	2	510	47	88.18
3	330	39	19.72	3	480	48	91.38
4	360	43	19.73	4	510	52	89.68
5	310	40	19.64	5	450	48	85.91
6	360	41	19.04	6	510	49	86.12

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
