# Peer review of "Analytical Modeling of In-Process Temperature in Powder Bed Additive Manufacturing Considering Laser Power Absorption, Latent Heat, Scanning Strategy, and Powder Packing"

_materials, 2019, doi:10.3390/ma12050808_

Reviewer 1 Report

In this work, the authors present an analytical model of the in-process temperature distribution in PDAM considering several factors. Overall the paper is well-written, but it very short and lacking a lot of detail normally seen from similar papers. Technically, the paper seems sounds and the results are reasonable from my experience - however, I have some serious concerns with the paper overall and cannot recommend further consideration of the paper for publication until these things are fully addressed. Please note that none of my comments are aimed at the authors personally nor should be taken as criticism of their skills or research professionalism. 

Overall: 

As previously stated, I have no complaints about the work from a technical perspective - but more details should be given about the model and its setup. The length of the paper and the amount of detail presented are appropriate for a technical brief but not a full article. There is very little discussion of the results and very poor detail about the experiments themselves.  

Major points, background and intellectual contribution:

There are at least dozens of very similar papers on MPBAM - more than enough for its own in-depth review paper. Good quality process models have existed for these AM processes since at least 2001. With this in mind, the background review is very poor and provides very little of the context and environment that similar previous work was done under. 

This is the fourth SLM/DMLS process model paper I have reviewed in the past 6 months (for several journals) and am very familiar with the literature on this topic - and I am not convinced that there is enough new/novel material in this paper to justify publication of a new paper in an academic journal even if the authors addressed the other presentation issues. Perhaps there are new and interesting contributions, but the authors have not shown it sufficiently to convince me that this project gives new or novel insights or solutions for the problem. I think that a much more thorough background review is needed before the authors could show this, so re-doing the literature review would be a good first step to addressing this. I expect that a rigorous review to show the value of a new MPBAM model would review a minimum of 100 references, perhaps more. 

The authors mention the residual stress problem in SLM/DMLS as a major influence on the problem and any solution, but discuss it very little and only provide one secondary reference discussing it. There are at least 100 good papers on this topic, in addition to several good review papers that the authors could refer to. 

The excellent SLM modeling work by the Kruth group at KU Leuven (reviewer is not affiliated with them in any way) is mostly ignored in this work, which should certainly be addressed. They have at least 30 related papers published and their work should be examined carefully, including several good papers on MPBAM residual stresses. NASA, Oak Ridge National Lab, and Lawrence Livermore National Lab all have excellent work that should be carefully reviewed before presenting a new MPBAM model.  

Why are there at least seven self-citations out of the 33 references, especially since (References 16-22) none of them seem to actually directly relate to this study or the background discussion of the work? I am familiar with the background research from the lab where this research came from and a lot of good work by some of the same authors is ignored. Some self-citation is unavoidable but it should be related only to the study directly to avoid any appearance of citation stacking by the authors.  

Finally, it appears that Author 1 did all of the work related to this paper with only "general guidance" from Author 2 and Author 4. It is not clear what Author 3 contributed to this paper. The authors need to more carefully specify what the intellectual contribution of each author is or remove all of the authors who did not actively participate in the research. "General guidance" and funding should be stated in the acknowledgements section and is not a criteria for authorship without some major contribution to the research. 

Major points, specific comments: 

Line 202: "Trial and error method"? Is this a formal method or just the authors guessing? Why not use laser absorption values from the literature? 

Line 203-207: There is not nearly enough detail given about the computer or the approach taken for the reviewer to understand what what done to reduce the computational cost or if the computational time is reasonable. 

Figure 3: It is not clear if this figure refers to simulation results or experimental work. I highly doubt that the absorption from [0,40%] follows a linear pattern. Much more detail is needed. 

Finally, some minor points: 

The abstract is way too long (should be 200 words maximum)

Use standard MPBAM keywords for the paper

Author Response

The authors appreciate the reviewer’s efforts on improving our manuscript. A detailed response to the reviewer’s comments is attached.

Reviewer 2 Report

The authors build an analytical model to investigate the in-process temperature in PBF, which is used to determine the part dimensions in Inconel 625. The authors use a stationary frame of reference which they claim, allows them to investigate molt pool evolution, growth and stabilization. The author validates the model with experimental data from ref 31. The idea is great, however, there are a few there are a few concerns which requires explanation and some suggestions to further improve the scientific value of the manuscript. I ask that authors specifically address each of my comments in their response and update the manuscript accordingly.

Comments:

 1.       It is very important to discuss the advantage and shortcomings of the analytical model for clarity. In this regard, it is not entirely clear to me how the analytical model takes in to account the typical multi-layer aspect of additive manufacturing process. This is of great importance because the thermal interactions between successive layers affect the temperature gradients, which govern the heat transfer and thermal stress development mechanisms.

 2.       The authors claim that the fixed origin modeling helps in calculating the part dimensions. However, it is not reflected in any of the subsequent results/analysis? There is a discussion regarding molten pool dimensions. How does this analysis translate to an actual part with finite dimensions (x, y, z)?

 3.       Eq (1) is typically used to investigate the moving heat source for thin-plates. It has some additional assumptions such as semi-infinite dimension, no hear generation, and no surface losses etc. There are two questions:

 a.       Does author assume single bead or multiple bead tests equivalent to thin plates? How did authors account for the build plate thickness in the model?

b.       I may be missing something, but the energy balance equation is not discussed in the manuscript?

 4.         The transformation from Eq (4) to Eq (5) is not clear. Please elaborate.

 5.       The authors analyze the melt-pool length and depth in this manuscript – is there a specific reason why the melt-pool width not analyzed? Based on the experimental data presented in ref 31, melt-pool width is most sensitive and shows larger variation with change in process parameters. Since the analytical solution requires very little time to simulate, the authors are highly recommended to include melt-pool width calculation/simulation to increase the scientific value of the manuscript. Moreover, molten-pool width is an important parameter in determining the part dimensions which is one of the claims of the manuscript.

 6.       The measurement and analysis of melt-pool length and depth in single-track and bidirectional-track are very unclear. Specifically:

a.       Please include the layer thickness, power size distribution to bring clarity while interpreting the data.

b.       Are the dimensions in Table 4 based on the data during stabilized phase? Can authors identify the stabilized phases in terms of coordinates (x, y) – where this phase is occurring.  E.g., For Figure 5, can you translate the x-axis from scan time to distance - this would give readers a perspective to when the simulation is stabilized. Provide similar plot for the bidirectional scan.

 c.       In ref 31, it is discussed that the melt-pool dimensions continuously change in the case of multi-track scans because of the presence of heat affected zone from the previous track. Specifically, the 1st and 3rd layer show significantly larger melt-pool dimensions compared to 2nd layer in a bidirectional scan. Could authors comment on this statement and how it ties up with the author’s findings?

d.       What is the shrinkage in the melt pool after solidification? Did authors compare (with experimental data) the melt-pool length and width before solidification or after solidification? It would be interesting to discuss the shrinkage in the melt-pool.

7.       The authors mention in the abstract that that residual stress can be predicted which this model. As a part of discussion, can authors elaborate on how this can be accomplished?

Author Response

(The authors gave the same response as above.)

Reviewer 3 Report

The paper “Analytical modeling of in-process temperature in powder bed additive manufacturing considering laser power absorption, latent heat, scanning strategy and powder packing” by J. Ning et al. deals with the use of an analytical solution of the heat diffusion equation for a moving point heat source and its application in predicting the thermal profile induced by laser power adsorption in an Inconel 625 sample.

The paper has some interesting aspects but its content largely derives from other studies proposed in literature. My critical observations are as follows:

1)      The analytical approach adopted in chapter 2 refers to an infinite medium, as amply discussed in the book of Carslaw and Jaeger (cited in the references), but the authors have applied this formalism to model heat diffusion from a moving pointlike source in a finite medium. A more correct approach would rely upon the use of the proper Green function for a bounded domain, as suggested in the same reference book. The author should motivate their ansatz with a deeper analysis.

2)      The authors claim that the laser absorption (40%) has been estimated by experimental data. The temperature profiles of Fig.2 have been calculated accordingly. However, the temperature values reported in Fig.6 do not seem to match with the ones typical of Fig.2 and Fig.4. Why?

3)      Chapter 3 is characterized by a somewhat confusing way of presenting the results. Namely, there is a persistent ambiguity between experimental and predicted data. Please clarify whether the graphs are referred to experimental or simulated data both in the text and in the captions of the figures.

4)      The diffusion of heat in porous media cannot be accurately investigated by simply adding a dependence of density and diffusivity on porosity. Many aspects, typical of diffusion in disordered media, have been overlooked with superficiality. A discussion should be added explaining the choice of this simplified and coarse model to study heat diffusion in porous media.

Minor observations:

5)      Many symbols and constants in Chapter 2 are left undefined. The same applies to some acronyms in the Introduction. Please check and explain.

6)      There are many grammar and syntax errors sparse in the text.  Examples: lines 223-224, etc…

Author Response

The authors appreciate the reviewer’s efforts on improving our manuscript. A detailed response to the reviewer’s comments is attached.

Round  2

Reviewer 1 Report

The authors have done an impressively extensive and fast revision of the paper. They addressed all of my comments and I have no further concerns about the manuscript. 

Author Response

The authors would like to express their sincere appreciations again to the reviewer for helpful comments and suggestions. Thank you.

Reviewer 2 Report

The authors have successfully addresses my comments and I have no further concerns.

Author Response

(The authors gave the same response as above.)

Reviewer 3 Report

The author have given a satisfactory answer to the questions of this reviewer and they have improved the overall quality of the present paper.

As a final recommendation, I suggest to specify, in the text or in the captions of the figures, whether all simulated plots refer to the same adsorption fraction (0.4) typical of Fig.2.

After the authors have made this clarification, this paper can be accepted.

Author Response

Clarification of the absorption has been made in the text and the captions of figures. Please find the details in the attachment.

The authors would like to express their sincere appreciations again to the reviewer for helpful comments and suggestions. Thank you.
